



# Stable isotopic constraints on global soil organic carbon turnover

Chao Wang[1], Benjamin Z. Houlton[2], Dongwei Liu[1], Jianfeng Hou[1,3], Weixin Cheng[1,4], Edith Bai[1,5,*]

1CAS Key Laboratory of Forest Ecology and Management, Institute of Applied Ecology, Chinese Academy of Sciences, Shenyang, 110164, China.

2 Department of Land, Air and Water Resources, University of California, Davis, CA, 95616, USA.

5    3College of Resources and Environment, University of Chinese Academy of Sciences, Beijing, 100049, China.

Department of Environmental Studies, University of California, Santa Cruz, 1156 High Street, Santa Cruz, CA, 95064, USA.

5School of Geographical Sciences, Northeast Normal University, Changchun, 130024, China.

*Correspondence to: Edith Bai (baie@iae.ac.cn)*

**Abstract.** Carbon dioxide release during soil organic carbon (SOC) turnover is a pivotal component of atmospheric $CO_2$
concentrations and global climate change. However, reliably measuring SOC turnover rates at large spatial and temporal scales
remains challenging. Here we use a natural carbon isotope approach, defined as beta (β), which was quantified from the $\delta^{13}C$ of
vegetation and soil reported in the literature (182 separate soil profiles), to examine large-scale controls of climate, soil physical
properties and nutrients over patterns of SOC turnover across terrestrial biomes worldwide. We report a significant relationship
between β and calculated soil C turnover rates ($k$), which were estimated by dividing soil heterotrophic respiration by SOC pools.
ln(-β) exhibits a significant linear relationship with mean annual temperature, but a more complex polynomial relationship with
mean annual precipitation, implying strong-feedbacks of SOC turnover to climate changes. Soil nitrogen (N) and clay content
correlate strongly and positively with ln(-β), revealing the additional influence of nutrients and physical soil properties on SOC
decomposition rates. Furthermore, a strong ($R^2 = 0.85$; p<0.001) linear relationship between ln(-β) and estimates of litter and root
decomposition rates suggests similar controls over rates of organic matter decay among the generalized soil C stocks. Overall,
these findings demonstrate the utility of soil $\delta^{13}C$ for independently benchmarking global models of soil C turnover and thereby
improving predictions of multiple global change influences over terrestrial C-climate feedback.

## 1 Introduction

Soil contains a large amount of organic carbon (C) and plays a crucial role in regulating Earth's C cycle and climate system
(Schmidt et al., 2011;Reichstein et al., 2013). Approximately 1500 Gt of soil organic carbon (SOC) is stored in the upper meter of
global mineral soil (Scharlemann et al., 2014), which is equivalent to ~160 years-worth of current fossil fuel $CO_2$ emissions.
Disagreement exists, however, over the residence time of this vulnerable C stock and its relationship to factors of ongoing change,
particularly climate changes and widespread nitrogen pollution (Reay et al., 2008;Reichstein et al., 2013). Biogeochemical models
rely heavily on turnover rates of discrete SOC pools (active, intermediate, and recalcitrant) derived from lab incubation studies
(Davidson and Janssens, 2006;Xu et al., 2016). In practice, however, SOC pools fall along a continuum of characteristic turnover
times (from days to centuries; (Schmidt et al., 2011;Lehmann and Kleber, 2015)), in a given ecosystem site. Furthermore, lab-
derived estimates of SOC turnover disrupts the sensitive balance between plant-soil-microbe interactions in ecosystems, adding
questions on the reliability of such techniques when applied to real-world conditions.

The rate of SOC turnover is an important parameter for process-based ecosystem models (Davidson and Janssens,
2006;Schimel et al., 1994) and those used to forecast the global carbon cycle and climate system in the future (Friedlingstein et al.,
2006). Global biogeochemical models often use climatic factors such as precipitation and temperature to  predict SOC turnover
rates (Schimel et al., 1994;Nishina et al., 2014). While several studies reported positive relationships between temperature and
SOC turnover (Chen et al., 2013;Trumbore et al., 1996;Bird et al., 1996;Trumbore, 1993;Carvalhais et al., 2014), however, others



have called the generality of such relationships into question (Giardina and Ryan, 2000). This discrepancy could be due to interactions among factors which are difficult to separate in the field, for example, among soil temperature, soil moisture and

nutrient controls over SOC decomposition (Davidson and Janssens, 2006). Nitrogen (N) in particular can affect SOC decomposition by changing microbial community structure, microbial activity or both (Curiel et al., 2007). Incorporation of factors besides climate is crucial for improving model performance and predicting the feedback-response of the terrestrial carbon cycle to climate change (Nishina et al., 2014).

        In addition, questions remain regarding whether the turnover of different C stocks behaves fundamentally similarly. For

example, climate is considered to be a "master regulator" of leaf litter (Zhang et al., 2008), root (Gill and Jackson, 2000;Silver and Miya, 2001) and soil organic C pools (Davidson and Janssens, 2006). However, recent findings have pointed out that soil microbial community composition may play a more important role in litter decomposition rates than climate or litter quality (Bradford et al., 2016;Keiser and Bradford, 2017). Due to the different microbial communities among leaves, roots and soils, and different chemical composition of such pools, rates of C turnover have the potential to vary widely across generalized classes of C stocks.

Stable carbon isotope composition ($\delta^{13}$C) can provide critical and relatively non-disruptive insights into the turnover of SOC (Garten et al., 2000;Accoe et al., 2002;Powers and Schlesinger, 2002;Bird et al., 1996). For sites with reasonably stable vegetation stocks, measures of vertical soil-profile $\delta^{13}$C can provide constraints on SOC turnover rates in ecosystems (Acton et al., 2013;Garten et al., 2000;Wynn et al., 2006). Soil $\delta^{13}$C generally increases from shallow to deep mineral soils in relatively well-drained systems, concomitant with decreasing SOC concentrations (Fig. S1). The vertical distribution of the $\delta^{13}$C reflects microbial

preferences for $^{12}$C vs. $^{13}$C in decomposing substrates (Garten et al., 2000), which, in turn, increases the $^{13}$C/$^{12}$C of residual organic C fractions with a kinetic isotope effect defined by ε (Fig. S1). Therefore, SOC $\delta^{13}$C tends to increase with depth along vertical soil profiles until it reaches a maximum value at which point a steady-state is achieved (Kohl et al., 2015;Accoe et al., 2002;Brunn et al., 2014;Garten et al., 2000;Wynn et al., 2006;Brunn et al., 2016). These trends result in a negative linear relationship between the log-transformed SOC concentration and soil $\delta^{13}$C (Acton et al., 2013;Garten et al., 2000;Garten and Hanson, 2006;Powers and

Schlesinger, 2002). The slope of the linear regression between soil $\delta^{13}$C and the log-transformed SOC concentration is defined as beta (β), which has been proposed as a proxy for SOC turnover rate in a select number of sites (Acton et al., 2013;Garten et al., 2000;Powers and Schlesinger, 2002). β has also been assessed in set of regional scale analyses (Acton et al., 2013;Brunn et al., 2014); however, whether β values can be used to constrain rates and controls on SOC turnover is yet to be explored at the global scale.

Here, we examine the efficacy of β as a proxy for SOC turnover rates by synthesizing soil profile data from sites around the world (Fig. 1). To understand the overall utility of C isotope composition for constraining SOC turnover rates, we explore the relationship between β and modeled SOC decomposition constant *k* and environmental factors, particularly climate, soil clay content and nutrient availability. We also compare the variation of β with that of root and litter turnover rates across latitude (thermal) gradient to examine whether and how the decomposition of generalized C pools varies as a function of likely controls.

**2 Materials and Methods**

**2.1 Data compilation**

Using the key words of 'carbon isotope & vertical profile', '$\delta^{13}$C & soil depth profile', or 'soil carbon turnover & stable isotope' on the Web of Science source, we assembled a total of 155 soil profiles from 50 journal papers (Fig. 1; A list of the literature sources is given in Table S1). Only soil profiles under natural ecosystems (mainly C$_3$ vegetation) without significant human





disturbance were selected. For each profile, we collected carbon isotope ($\delta^{13}C$), organic carbon (SOC) and N concentration of leaf/litter and mineral soil layers at different depths if the data is available, and more than four $\delta^{13}C$ values should be provided within the top 1 meter. Where data were not available in tables, Data Thief software (http://www.datathief.org/) was used to acquire values from figures. We also noted the experiment location (latitude and longitude), biome types, mean annual precipitation (MAP), and mean annual temperature (MAT). In cases where climate variables were not reported, we used the WorldClim data

(http://www.worldclim.com/) to reconstruct climate values based on latitude and longitude coordinates in ArcGIS version 10.0 using Spatial Analysis tool (ESRI, Redlands, CA).

In addition, a previous reported arid and semi-arid grassland transect along 3000 km with 27 sampling locations was added into the dataset (Wang et al., 2017). Those sampling sites are dominated by $C_3$ plants and cover approximately 16° longitude ranging from 104°52′ E to 120°21′ E and 10° latitude ranging from 40°41′ N to 50°03′ N. The MAP ranges from 90 mm to 420

mm and MAT ranges from -2 ºC to +7 ºC.  At each location, five 1 m × 1 m sub-plots (or one 5 m ×5 m sub-plot in areas with shrub as the dominating plants) were setup within a 50 m × 50 m plot. Twenty soil cores (0 - 100 cm) in each 1 m × 1 m sub-plot were collected and divided into 0-10 cm, 10-20 cm, 20-40 cm, 40-60 cm and 60-100 cm depth segments and bulked to form one composite sample for each segment per sub-plot. Leaf samples of five dominating genera (*Stipa, Leymus, Caragana, Reaumuria* and *Nitraria*) were sampled for carbon isotope analysis if these genera were present in the sub-plots.

In laboratory, leaf samples were washed with deionized water to remove dust particles and then dried at 65 ºC for 48 h. Both soil and leaf samples were ground in a ball mill and stored in a plastic bag. Soil carbonate was removed from soil samples using 0.5 M HCl. Organic carbon concentration and isotope composition of soil and leaf were carried out at the Stable Isotope Faculty of University of California, Davis.

### 2.2 Beta calculation

A negative linear regression between the log-transformed SOC concentration and $\delta^{13}C$ for each soil depth profile was conducted (Fig. S1). The slope of the this linear regression is defined as beta  ($\beta$) value (Acton et al., 2013;Garten et al., 2000;Powers and Schlesinger, 2002).

### 2.3 Soil decomposition rate constant (*k*)

We assumed soil profiles in our dataset are at or near steady-state conditions and carbon decomposition rate constant (*k*) was

estimated as the ratio between soil heterotrophic respiration ($R_H$) and soil organic carbon stock (SOC) (Sanderman et al., 2003):

$k = R_H/ SOC$                        (1)

The SOC stock and the mean annual soil total respiration ($R_s$) of each profile (1 m depth), respectively, were extracted from global soil-property database (Zinke et al., 1998) and a climate-driven regression model(Raich et al., 2002). All data extraction was processed in ArcGIS version 10.0 using spatial Analysis tool (ESRI, Redlands, CA). Then, we used the linear relationship

between soil respiration ($R_S$) and $R_H$ to calculate $R_H$ (Bond‐Lamberty et al., 2004):

$\ln (R_H) = 1.22 + 0.73 \times \ln (R_S)$     (2)

### 2.4 Data analysis

Negative $\beta$ value and decomposition rate constant *k* were log-transformed to perform statistical tests. Larger $\ln(-\beta)$ translated to faster SOC decomposition rates (Acton et al., 2013;Powers and Schlesinger, 2002). Soil $\ln(-\beta)$ was analyzed and across different

biome types. Soil $\ln(-\beta)$ was also compared with litter and root decomposition rate along latitude at the global scale. Two-variable regression analysis was first performed to explore the relationship between $\ln(-\beta)$ and $\ln(k)$, or $\ln(-\beta)$ and climate variables (MAP



and MAT) as well as soil edaphic factors (N and clay content). Multiple regression analysis was then used to examine the relationship between ln(-β) and these variables (MAP, MAT, soil N and clay content). Akaike information criterion (AIC) was used to estimate the quality of model when increasing the number of parameters.

**3 Results**

**3.1 Worldwide patterns of β**

A total of 182 soil profiles from all continents other than Antarctica were encapsulated in our compiled dataset (Fig. 1). Carbon isotope composition ($\delta^{13}$C) increased with soil depth in the majority of profiles examined and was strongly correlated with the natural logarithm of SOC (Table S1). ln(-β) was significantly positively related with site-based estimates of the soil C

decomposition constant, ln($k$) with $R^2$ = 0.41 (Fig. 2).

The values for ln (-β) ranged from -0.50 to 2.20 across sites (non-transformed β values ranged from -0.60 to -9.10, Table S1). Highest mean ln(-β) was observed in tropical savanna regions (Fig. 3), followed by tropical forest, desert, temperate forest, and temperate grassland. MAP among those five biomes increased from desert < temperate grassland < temperate forest < tropical savanna < tropical forest and MAT increased from temperate grassland < temperate forest < desert < tropical forest < tropical

savanna (Fig. 3).

Along the latitude gradient, ln(-β) decreased from the equator to poles, but was higher at 20-30° N compared to the 10-20° N latitudinal band (Fig. 4a). The mean decomposition rate of leaf-litter and root C displayed similar latitudinal patterns ($R^2$ = 0.85; p<0.001, Fig. 4b).

**3.2 Controls on β across ecosystems**

ln(-β) and MAT displayed a strong, positive relationship across the global dataset ($R^2$ = 0.51; $P < 0.001$; Fig. 5a).  Ln(-β) did not show a simple linear correlation with MAP, but instead showed a polynomial relationship with a tipping point at MAP = 3000 mm (Fig. 5b). When MAP was less than 3000 mm, ln(-β) was positively correlated with MAP ($R^2$ = 0.37, $P < 0.001$, data not shown); ln(-β) decreased with the increasing of MAP in areas receiving > 3000 mm of MAP. A quadratic equation provided the best fit to the relationship between ln(-β) and MAP for all sites ($R^2$ = 0.31, $P < 0.001$; Fig. 5b). Soil N explained 25% of the variation in ln(-

β) ($P < 0.001$; Fig. 5c). Moreover, a quadratic equation best described the relationship between soil clay and ln(-β), with $R^2$= 0.28 ($P < 0.001$; Fig. 5d). AIC analysis showed that the full-factors model (i.e., MAT, MAP, soil N and clay) accounted for more of the variation in ln(-β) than any other regression model in the global data set (Table 1).

**4 Discussion**

Our global data synthesis reveals significant relationships between ln(-β) and the turnover of soil, litter and root C pools at

geographically broad scales (Fig. 2; Fig. 4). These findings build on site-based observations and regional assessments (Accoe et al., 2002;Garten et al., 2000;Powers and Schlesinger, 2002), and suggest that C isotope composition is a useful proxy for understanding generalized patterns of SOC turnover and the underlying controls over soil C metabolism. That our results link to all soil C pools implies that SOC, root and litter turnover share common controls, in particular those related to climate and nitrogen. These findings suggest that the decomposition processes of belowground and aboveground may have similar responses to global

climate change, such as global warming and increasing atmospheric N deposition. Furthermore, our results highlight the potential of incorporating natural stable C isotopes in global biogeochemical and Earth system models to constraint soil and litter decomposition rates that are vital to climate change forecasts.





Within the terrestrial biosphere, our findings point to highest mean ln(-β) in tropical savanna, implying rapid average SOC turnover in this expansive and dynamic biome (Fig. 3). Several previous studies have suggested that SOC decomposition rates are

highest among tropical rainforest (Carvalhais et al., 2014); however, tropical rainforest ranked second to savannas in our global synthesis (Fig. 3). This difference might be due to data coverage, moisture or nutrient effects, or all of these. Our estimates of ln(-β) were not area-weighted, nor did the data cover this diverse ecosystem exceptionally well; the small area of wet tropical sites in our analysis may therefore not reflect the average conditions of tropical lowland forest. The tropical forest data were largely from sites in Costa Rica and Brazil, with MAP equal to 4058 mm and 1359 mm, respectively. Our analysis thereby points to the need

for more data from an array of tropical forest sites, given the substantial biogeochemical diversity within this globally important biome (Townsend et al., 2008).

At the other extreme, our analysis suggests that slowest mean rates of SOC decomposition occur in temperate grassland (Fig. 3), consistent with results from previous simulation modeling (Carvalhais et al., 2014;Schimel et al., 1994). Relatively slow decomposition rates have been observed for plant litter decay in arid grassland sites (Zhang et al., 2008), and largely reflects strong

moisture controls on decomposition. In addition, microbial biomass and microbial activities are much lower in arid/semi-arid vs. mesic or humid sites (Fierer et al., 2009), thus leading to low rates of SOC and litter decomposition.

The differences of β value among different biomes reflects several controlling variables – especially mean annual temperature, mean annual precipitation, soil N contents, and clay content.  Of particular importance are temperature-driven controls over β, in which MAT explains ∼ 50% of the variation of ln(-β) in our global data compilation (Table 1; Fig. 5a). A recent meta-analysis,

which included 24 soil profiles across a range of cool temperate to tropical forest sites, reported similarly strong temperature-dependencies of β (Acton et al., 2013). Our findings broaden this perspective to a global range of terrestrial biomes and climates, and indicate that, with increasing MAT, SOC turnover is substantially accelerated. This result agrees with previous studies which have identified temperature as the strongest regulator of soil C decomposition among all known controls (Carvalhais et al., 2014;Schimel et al., 1994), and is consistent with global C-climate feedback models, which project acclereated rates of $CO_2$ efflux

from the land biosphere with climate warming (Ciais et al., 2014).

Our study also points to significant relationships between β and precipitation-climates, which are more complex than those observed for MAT. Rather, we find an inflection point in β in our global data set at MAP ∼ 3000 mm (Fig. 5b). This relationship reveals negative effects of moisture on SOC decomposition rates in very wet climates, which account for a small fraction of the terrestrial biosphere. Most of the C isotope data used to estimate β were available in sites with MAP < 3000 mm, which collectively

account for > 98% of the world's terrestrial ecosystem rainfall regimes. In extremely wet sites, it is likely that leaching of dissolved organic carbon (DOC) from soils to streams affects the relationship between decomposition and isotope effect expression (Powers and Schlesinger, 2002). Previous studies have shown that the $\delta^{13}C$ of DOC increases with increasing soil depth (Kaiser et al., 2001). Because DOC is generally $^{13}$C-enriched (Kaiser et al., 2001), increasing DOC leaching into very wet sites would be expected to induce a larger change in soil $\delta^{13}C$ with depth, and hence, increasing ln(-β). Thus, we may have overestimated the SOC turnover

rate in areas with high DOC leaching; however, this cannot explain the low ln(-β) values in areas with MAP > 3000 mm (Fig. 5b).

A cross-system compilation of the smaller though more dynamic litter pool shows a similar pattern of decreasing decay rates in regions with MAP > 3000 mm compared to drier sites (Zhang et al., 2008). In addition, Schuur (2001) showed that leaf and root decomposition rates declined significantly with increasing precipitation along a highly constrained rainfall sequence in Hawaiian forest sequence (from 2020 mm < MAP < 5050 mm), thereby resulting in slower rates of nutrient mineralization and declines in

net primary production (NPP) in the wettest sites. The consistencies between our study and past work suggests that precipitation affects the decomposition of SOC and litter in similar ways, slowing decomposition rates when MAP is very high and anaerobic conditions dominate (i.e. MAP > 3000 mm; Schuur 2001).



In addition to climate, nutrients influence the magnitude of ln(-β) in our compilation, with SOC turnover rates generally increasing with soil N concentrations across ecosystem sites (Table 1, Fig. 5c). Although soil N has been suggested as an important control over SOC decomposition in previous work (Schimel et al., 1994), our study is one of the few to confirm the existence of such a relationship at the global scale. Positive correlations between litter decomposition rates and litter N contents during the early stages of decay have been reported previously (Berg, 2000). Past work has also suggested that high N availability enhances soil degrading enzyme activities (Fioretto et al., 2007).

Finally, our results suggest that soil physical factors, particularly soil clay content, plays a role in ln(-β) and soil organic C turnover (Fig. 5d), consistent with previous expectations (Schimel et al., 1994;Xu et al., 2016). In sites where clay content is < 50% (i.e., sandy soils), for example, ln(-β) increases with the soil clay content; however, when clay content is > 50% (loamy or clayey soils), no clear relationship between ln(-β) and clay content is observed (Fig. 5d). The change in this relationship could be explained by the higher "preservation capacity" of clayey soils (Vogel et al., 2014).

SOC turnover is an important parameter for process-based models and Earth system models (Schimel et al., 1994;Davidson and Janssens, 2006), and models used to forecast the carbon cycle and climate system into the future (Friedlingstein et al., 2006). Global biogeochemical models commonly use climatic factors as predictors of SOC turnover rates (Carvalhais et al., 2014). In contrast, our results point to factors beyond climate singly, soil N content and soil texture, in altering organic C turnover across the terrestrial biosphere. Taken together, for instance, our multiple regression analysis considering all factors (i.e., MAT, MAP, soil N and clay) explains nearly 70% of variation of ln(-β) ($R^2$= 0.67, $P$<0.001; Table 1), suggesting the high dependence of SOC turnover on these factors. We therefore suggest the need for models that include all of these factors when forecasting global C cycle response to change.

In addition, our findings suggest that the C isotope composition of the soil can help to improve global C model performance. A common problem in global C research is finding consistent and sufficiently integrated metrics against which the performance of different biogeochemical models can be quantitatively analyzed (Tian et al., 2015). The strong relationships we observe between β and SOC turnover suggest that this natural-isotope proxy can be used to ground-truth large-scale patterns of model-simulated soil C dynamics. Future work to collect and analyze C isotope data in vertical soil profiles, which is a relatively inexpensive process, can further extend the regional coverage of β and help benchmark SOC turnover estimates among global model simulations. This is important given the potential for SOC turnover to respond to multiple global changes and produce significant feedbacks on climate at the global scale (Carvalhais et al., 2014;Lehmann and Kleber, 2015).

Like any approach, however, the use of β has its own inherent biases and uncertainties, especially those factors influencing profile-trends in soil $\delta^{13}$C beyond kinetic isotope fractionation during decomposition. For example, atmospheric $\delta^{13}CO_2$ has been decreasing since the industrial revolution owing to the combustion of $^{13}$C-depleted fossil fuel, which could lead to lower $\delta^{13}$C in surface soils vs. deeper horizon (Friedli et al., 1987). Fortunately, this effect is minor (i.e., 1.4 - 1.5‰) compared to the substantial variation of soil $\delta^{13}$C along depth profiles (3.5‰) in our dataset (Accoe et al., 2002). Based on a 100-year-old soil archive (i.e. soil collected before extensive fossil fuel $CO_2$ emissions) and modern samples collected from a common site in the Russian steppe, Torn *et al*.(2002) showed $\delta^{13}$C profiles of modern and pre-industrial soils were similar, indicating that fossil fuel emissions with lower $^{13}CO_2$ has not significantly contributed to the gradient of soil $\delta^{13}$C with depth.

Second, bioturbation and consequent mixing of C from different sources has the potential to alter soil $\delta^{13}$C profiles (Acton et al., 2013). As the $\delta^{13}$C of root material is generally higher than that of above ground biomass, such as leaves (Powers and Schlesinger, 2002), the isotopic composition of SOC at the soil surface may be lower than deep soils. In addition, because microbes, invertebrates, and other soil fauna are typically enriched in $\delta^{13}$C compared to source-substrates, biological migration and physical mixing of soils may alter relationships between soil C concentrations and $\delta^{13}$C (Wynn et al., 2006). Process-based modeling

suggested that increased physical mixing of soil can reduce the trend in soil $\delta^{13}C$ from shallow to deep horizons (Acton et al., 2013). Moreover, DOC leaching could affect $\beta$ value; DOC is generally $^{13}C$-enriched (Kaiser et al., 2001), especially in soil with
high clay content and metal ions, such as $Ca^{2+}$, $Al^{3+}$, and $Fe^{2+/3+}$. However, since DOC accounts for < 10% of NPP across different terrestrial biomes (Neff and Asner, 2001), its effect on bulk soil $\delta^{13}C$ would be relatively small globally, though it could increase in importance where DOC contributes a higher fraction of NPP. Finally, steady-state assumptions in our analysis of $\beta$ may not hold for all ecosystems, and could thereby limit the quantitative use of $\beta$ in highly disturbed sites or in ecosystems undergoing substantial state-changes.

**5 Conclusion**

Our analysis provides a globally integrative tool for understanding variations of SOC turnover rate, which can be applied spatially based on estimates of factors such as climate and soil properties. Compared with other methods, utilization of C isotope composition ratios in soil profile provides an independent approach that does not rely on disruption of plant-soil-microbe interactions. It has the added benefit of integrating over longer time scales (decade to centuries), and thus provides a common
measurement for model-based benchmarking and calibration schemes.

**Author contribution.** C.W. and E.B. conceived and wrote the paper with contributions from B.Z.H. C. W., W. C., D. L., and J. H. conducted the field and laboratory works. C.W. E.B. and D.W.L. compiled data from peer-reviewed publications and conducted the modelling. All co-authors interpreted the results

**Acknowledgements.** This work was financially supported by the National Basic Research Program of China (973 program;
2014CB954400), the National Natural Science Foundation of China (31522010 and 41601255), and the Key Research Program of Frontier Sciences, CAS (QYZDB-SSWDQC006).

**Competing interests:** The authors declare that they have no conflict of interest

**Supporting information**

**Figure S1** Carbon isotope and concentration along vertical depth profile.
**Table S1** Database for $\beta$ analysis in this study.



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





**Table 1:** Multiple regressions between ln(-β) and climate and other edaphic factors at global scale.

| Variables | $R^2$ | $n$ | AIC |
|---|---|---|---|
| ln(-β) = 0.062 MAT -0.0001 MAP +0.204 | 0.55[***] | 182 | 619.10 |
| ln(-β) = 0.052 MAT -0.0001 MAP + 0.561 N + 0.169 | 0.59[***] | 105 | 249.55 |
| ln(-β) = 0.042 MAT -0.0001 MAP + 0.629 N + 0.003 Clay +0.100 | 0.67[***] | 74 | 140.04 |

MAT: Mean Annual Temperature (℃); MAP: Mean Annual Precipitation (mm); N: Soil nitrogen concentration (%). Clay: Soil clay concentration (%). $n$ is the number of data, and $R^2$ is the coefficient of determination for the regression line. AIC: Akaike information criterion. [***]represents significant at $p$ less than 0.001.






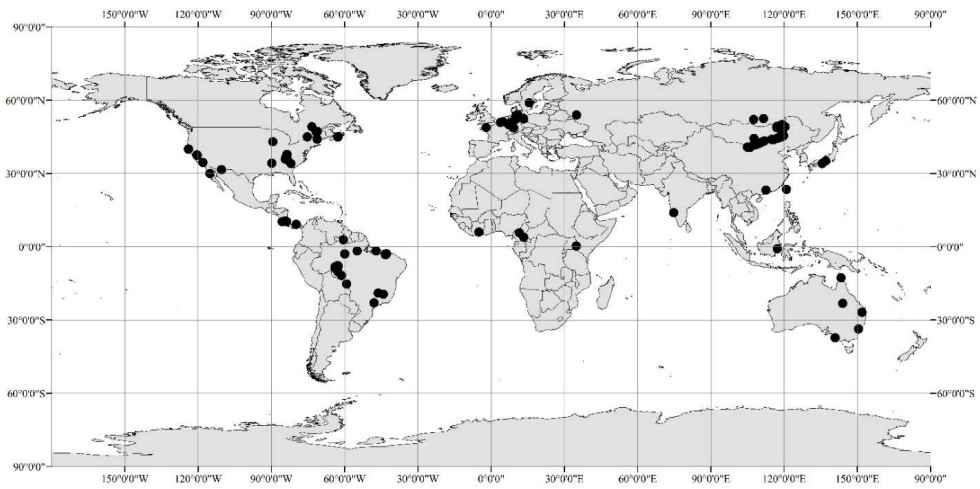

**Figure 1.** Locations of the 182 soil profiles used to calculate β values in this study.






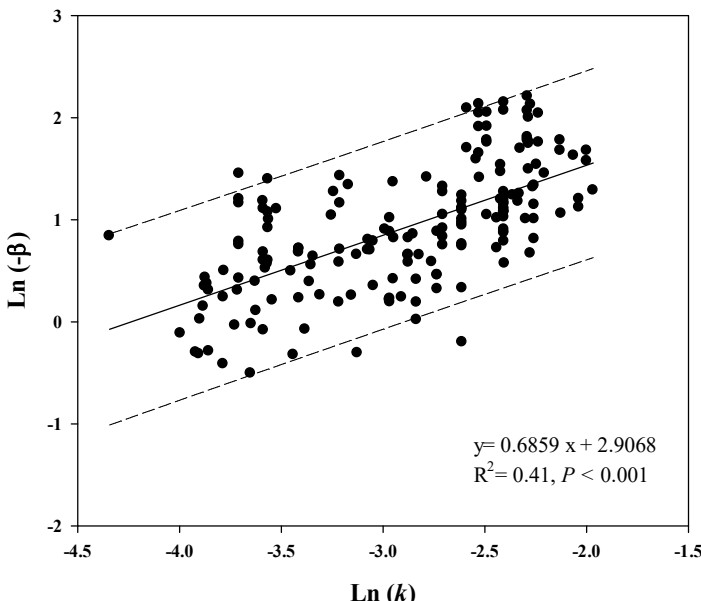

**Figure 2.** Link between β value and modeled soil carbon turnover rate ($k$), which was estimated as the ratio between soil heterotrophic respiration and soil carbon stocks. Solid line is regression line and dashed lines denote 95% prediction interval.






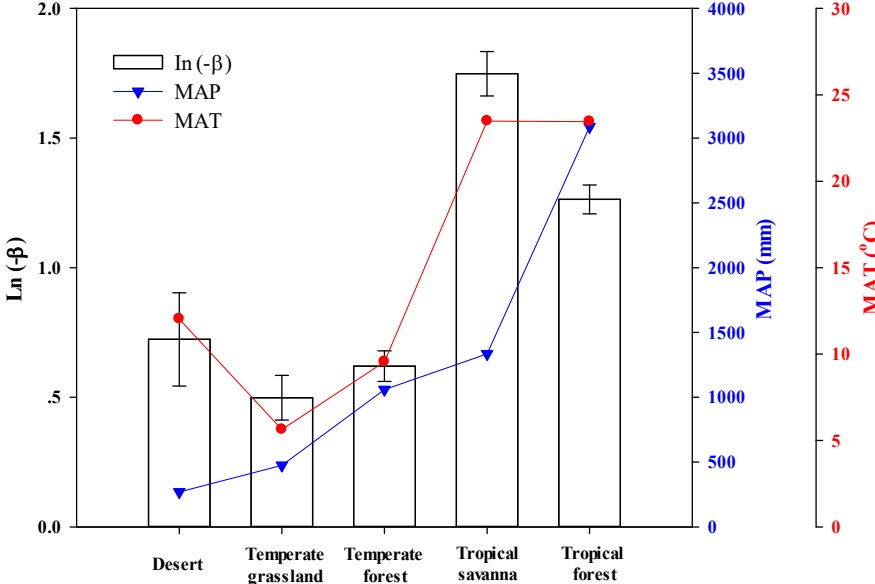

**Figure 3.** Variations of mean β with biome types. Blue and red points present MAP and MAT for each biome, respectively.



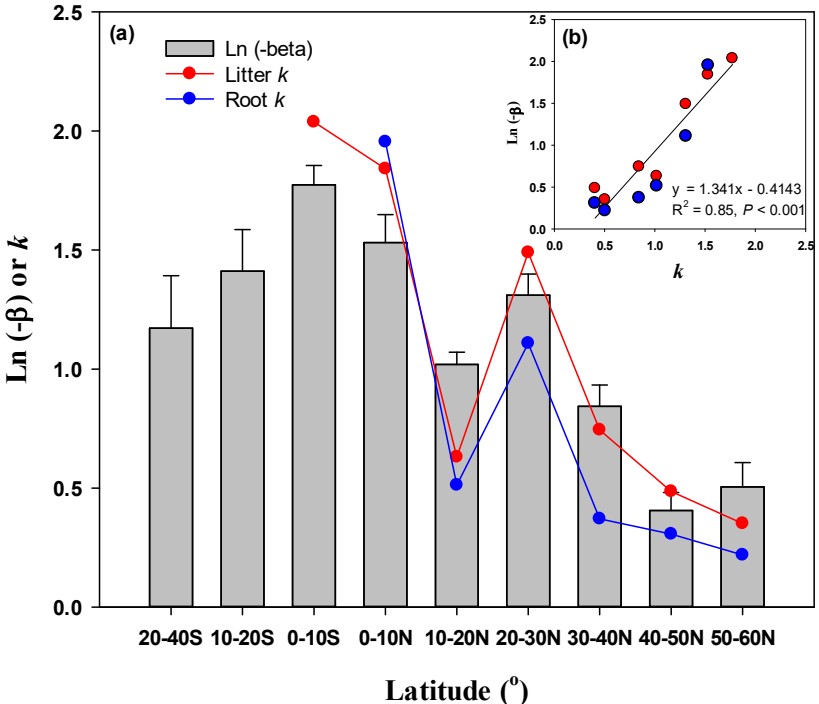

**Figure 4.** Variation of mean β value across latitude (bar chart), litter decomposition rate $k$ (yr[-1], red dots, Zhang et al., 2008) and root decomposition rate $k$ (yr[-1], blue dots, Silver and Miya, 2001) at the global scale. The inner panel is the regression between soil β value and litter and root decomposition rate $k$.



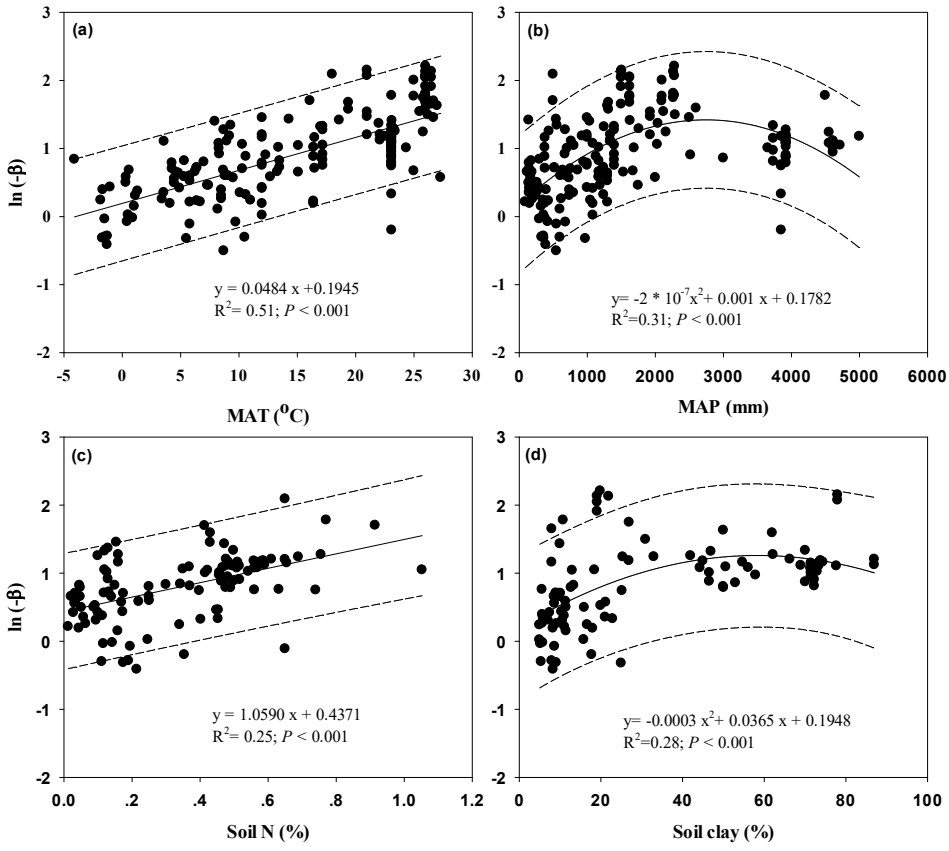

**Figure 5.** Beta varied with climate and edaphic factors. Relationships between ln(-β) and MAT (a), MAP (b), soil N
(c), and clay concentration (d) for global dataset. Solid line is regression line and dashed lines denote 95% prediction
interval.