# Peer review of "Stable isotopic constraints on global soil organic carbon turnover"

_Biogeosciences, 2017_

## Referee Comment (RC1) · J. Balesdent (Referee) · 24 Oct 2017

A. SUMMARY OF THE REVIEW The topic is fully relevent for publication in Biogeoscience. The study confirms the correlations between 13C enrichment down the depth in soil profiles and environmental variables, that bave been already reported by several authors on smaller datasets. The title makes sense. But the manuscript requires significant revisions for two major reasons. - The first concern the consistency of the dataset itself, i.e., some erroneous attribution of data to biomes, omission of true tropical savannas profiles and profiles that reveal paleo C4-vegetations. The contribution of past C4 vegetation (delta13C -10 to 13 ‰ in such profiles is misunderstood. This leads to (i) non representative values of beta and (ii) overestimation of temperature dependence of 13C enrichments. - The second point concerns interpretation beyond

this bias. The discussion tends to maximize the importance of kinetic fractionation by microbes as an explanation of beta's variance, but without providing results supporting it. I suggest reducing the section of discussion on this point. These comments are detailed in point "C. Extended review". Finally, the "Stable isotopic constraints on global soil organic carbon turnover " is there, but has to be recalculated after database correction. Interpretation of the correlations in terms of processes should be minored. I suggest to correct the database by restrict it to pure C3 ecosystems, and furthermore make the proposition to merge this database with another one, therefore doubling the number of observations (see B. below).

B. ONE UNUSUAL PROPOSITION TO IMPROVE THIS STUDY. In line with the philosophy of Biogeoscience Discussion, which stimulates interactive and cooperative research more than competitive research, I make the proposition to provide 155 additional profiles worldwide. In the frame of the COST (European COoperation in Science and Technology) action SIBAE (Stable Isotopes in Biosphere-Atmosphere-Earth System Research; 2009-2013), a group of 10 scientists (10 institutions) has built and analyzed an exactly similar database of 196 World 13C/C profiles and beta values under pure C3 ecosystems. 43 are common to yours, 99 in other peer-reviewed articles, and 56 in non peer reviewed literature (Figure 1). This dataset shows significant multiple regressions with climate and clay, which are similar to those presented in the present manuscript BGD 2017-338, but with less predictive value. This less predictive value is in accordance with the above-mentioned biases, with more profiles exhibiting less negative beta values (including some positive) and more varied environments. If the authors of MS BG2017-338 accept this proposition, a final dataset merging the present dataset (after correction) with SIBAE's one, would provide a stronger view of the stable isotopic constraints on global soil organic carbon turnover.

C. EXTENDED REVIEW.

1. Consistency of the database

Almost all so-called "tropical savannas" profiles in the database refer to afforestation or tree encroachments in C4 savannas. In published "real" tropical savannas profiles, delta13C decreases with depth, with values close to -12 -15 ‰ in surface and lower than - 20‰ in deep layers because of the presence of millenary-old forest-derived carbon. These have therefore POSITIVE beta values (of course non log transformable). If not log transformed, they would draw the "tropical beta" toward the opposite direction. In the so-called "tropical savannas" (profiles # 85, 86, 87, 106-112), reforestation leads to a strong gradient from C3 signature in the top, and predominant C4 signature below. These systems were precisely chosen by the authors to analyse C dynamics through 13C signature change, and are not representative of world savannas. In some of the cited papers, profiles with positive beta were omitted. Interpretations of highest ln(-beta) in this "expansive and dynamic biome" (line 149) are therefore based on forest expansion data! As a result Figure 3 is wrong: either real savannas should be included (positive beta) or tropical savannas and these C4 to C3 conversions should be removed from the database. The latter is my suggestion.

Beyond the case of "savannas", several profiles under tropical forests are marked by ancient C4 vegetations (profile numbers 78, 82-84, 115-121). They have been studied for this reason and are therefore not representative of world tropical forests. In both cases, beta is not linked to C turnover, but paleoclimate, as in many regions of the world. Almost all profiles with beta < -5 in the database are concerned. This overestimation of tropical ln(-beta) strongly affects Figure 4, the correlation with MAT or MAP (Figure 5), and Table 1, i.e., the main results.

2. Discussion of the relationships between beta and other variables (beyond paleo C4 vegetation) The discussion might sometimes be confusing. A "kinetic isotope fractionation" associated to biodegradation process (decay) would not directly imply a dependence on the rate (speed) of decay, i.e., the turnover rate. In the Rayleigh distillation equation, beta is typically independent on the rate. A partial explanation of the variance in beta by the turnover rate rely on complex processes (e.g., Acton,

Garten, Schlesinger), and should not neglect other sources of 13C variations, such as the change in plant isotopic composition with time, post-photosynthetic fractionation in plants, bioturbation, isotope composition of nitrogenous compounds, etc., which can also be involved in the correlation of beta with carbon turnover rate. On the contrary, the discussion tends to minimize these processes (lines 215 to 234). Furthermore, the magnitude of a kinetic fractionation by heterotrophic respiration in soils is still debated (e.g., Breecker et al., 2015). Since the results provide no new demonstration, I suggest minoring this part of the discussion. Since the dataset includes turnover rate (k), some hypotheses of factors affecting the "turnover rate" (e.g., MAP > 3000 mm) might be discussed also on the basis of k, and not only ln(-beta). The apparent decrease of ln(-beta) under climates with MAP > 3000 mm is probably linked to overestimated -beta in moderately moist tropical areas (C4 zone).

3. Details. Precise when defining beta that you used the decimal logarithm ("log" is ambiguous). Table 1 and Figure 5 legend: indicate that N(%) and Clay (%) refer to topsoil. Table 1 Add one digit to the regressor of MAP. Figure S1 is after Acton et al. 2013

Breecker, D. O. et al. 2015. Minor stable carbon isotope fractionation between respired carbon dioxide and bulk soil organic matter during laboratory incubation of topsoil BIOGEOCHEMISTRY 123, 83-98.

[Figure]

[Figure]

In each profile *beta* is defined as the slope of the logarithmic regression of SOM $\delta^{13}$C (‰ VPDB) vs carbon concentration (C in g. kg$^{-1}$) according to:

$$\delta^{13}C = a + beta.\log_{10}(C)$$

a) Location of profiles. b) Relationship between *bet*a and MAT, Annual Precipitations and topsoil clay content. Vertical bars stand for one standard deviation of *beta* as determined by profile individual regression.. After Balesdent J., Ågren G., Braakhekke M., Chadoeuf J., Derrien D., Gessler A., Hatté C., Kayler Z., Kuzyakov Y., and Wynn J. SIBAE Working group report. The carbon stable isotopes composition of organic matter down soil depth : a meta analysis. .

**Fig. 1.** Relationship between world soil organic matter delta13C gradient down the depth and environmental variables over 196 published soil profiles under C3 vegetation.

---

## Referee Comment (RC2) · Anonymous Referee #1 · 3 Nov 2017

The manuscript "Stable isotopic constraints on global soil organic carbon turnover" by Wang et al. presents an interesting approach of deriving information about SOC decomposition kinetics from stable carbon isotope information along the soil profile. For this, they derive a slope "beta" from the relationship of $\delta$13C values and SOC content of soil profiles across the globe, and then relate these "beta" values to calculated decomposition kinetic constants "k" (or more precisely their log-transformed negative values). They state that the highly significant linear relationship of the log-transformed variables can be used to derive SOC decomposition kinetics from $\delta$13C profiles of SOC. Furthermore, they relate these ln(-beta) values to four different parameters, i.e. MAT, MAP, soil clay and nitrogen content. For all four parameters they find significant relationships with ln(-beta). This approach is interesting and might be promising if proven to be reliable.

[Figure]

The weak part is the calculation of the kinetic decomposition constants with several secondary data sources and a fixed relationship between heterotrophic and total soil respiration, which might be too much of a simplification for this global approach, given the large range of ratios between heterotrophic and autotrophic respiration found for different ecosystems and conditions.

More specifically, the concerns are the following:

1) The kinetic decomposition constants k for the different soil profiles have been calculated by assuming steady-state conditions, i.e. SOC input and output are in equilibrium. While this assumption might hold true for many of the sites, there is no evidence provided that this really is the case.

2) The SOC stocks, which represent the denominator in equation 1, were extracted from the Global Organic Soil Carbon and Nitrogen (Zinke et al. 1998). There is no mention whether there was an exact match between the soil profiles used in the present study, or whether spatial approximations were made, and if yes, which criteria were used for these spatial approximations.

3) Heterotrophic soil respiration was calculated from total soil respiration by a fixed linear relationship adopted from Bond-Lamberty et al. (2004). Given the large variability of the fraction of Rh to total soil respiration (varying between 10% and 90% in vegetated ecosystems), this approach is highly questionable.

4) Also total soil respiration was not measured, but derived from a climate-driven regression model (Raich et al. 2002).

5) And finally, climate data were derived from WorldClim as a function of latitude and longitude (what about altitude?), whenever climate data were not available in the literature tapped in this study. Again, there is no mention whether there was an exact match between the locations of the present study, or whether spatial approximations were made, and if yes, which criteria were used for these spatial approximations.

Given all above-mentioned uncertainties concerning the calculation of the key variable of the study, i.e., the SOC decomposition rate constant k – which by the way is an apparent constant, as it is a composite of the decomposition of several SOC pools with different decomposability/recalcitrance – the reader would expect an extensive uncertainty analysis. However, not a single attempt was made to quantify those uncertainties, which certainly will amount to a large relative error due to multiple convolutions of single functions and error propagation. Also no mention is made of this crucial point in the discussion, and how this might affect the far-reaching conclusions drawn.

Therefore, I would have great concerns recommending acceptance of the paper in Biogeosciences, even after major revisions, as those concerns are aroused by the intrinsic weaknesses of the data sources, i.e. the dearth of own measured data of key components of the assessment presented here, which cannot be healed by a major revision.

---

## Author Comment (AC1) · 30 Nov 2017

Authors' response: We would like to thank Prof. J. Balesdent for his time and effort for reviewing this manuscript. We agree that there are some erroneous attribution of data to biomes, omission of true tropical savannas profiles and profiles that reveal paleo C4-vegetations. We will revise our dataset, correct the errors and remove soil profiles in tropical savanna ecosystems and those tropical forest soil profiles which might experience C4 vegetation changes according to the reviewer's comments. Hence, 26 soil profiles will be removed from our analysis. To compensate, we will add 20 new soil profiles into the revised dataset from 8 peer-reviewed articles, which will result in 176 separate soil profiles in the revised dataset-these studies were not available when we first started gathering our data set. Our preliminary review of these additional data will

not alter our summary conclusions, yet will add robustness to our overall global findings. We thank the reviewer's suggestion to add 20 more profiles from an unpublished source; however, because our analysis is focused on synthesis of the published literature, we would not add this dataset, but would like to share our full dataset once it is published.
* * *

---

## Author Comment (AC2) · 30 Nov 2017

Authors' response: We would like to thank reviewer for the time and effort put into reviewing our manuscript. We agree with reviewer's comments that there are uncertainties associated with kinetic decomposition constant k – which have been discussed previously. However, to our knowledge, the approach we took is the best available for providing integrative soil carbon decomposition rates estimates along profiles. Numerous published modeling studies have used approaches involving multiple data sources and assumptions – similar to our approach. While we agree that the coupling of different data sources inherently injects uncertainties – and we are happy to discuss the caveats in revision. Getting quantitative with the uncertainty is unfortunately not possible (as other studies have noted), for the following reasons.

[Figure]

First, we mainly focus on soil 13C-based proxy and its variations with MAT and MAP in this manuscript. Hence, we compared our beta value with the kinetic decomposition constant k to explore correlation between these two factors, but we can't quantitative assess the relationships at the global scale. This does not devalue the correlations we find across sites, though it does limit our quantitative assessment, pointing to an area for future research for the community. Thus, if useful, we could move the comparison between beta and k to the supporting information if this seems like the best approach.

Second, just because there are large uncertainties with the kinetic decomposition constant k, we believe the independent approach provided in this manuscript using carbon isotope variations along soil profiles is a promising. Indeed, it allows for larger-scale geographic exploration of soil carbon decomposition at the global scale in a way that differs fundamentally from current approach. We believe that this constraint can be used to help benchmark global models, which are lacking in their ability to generation global soil C patterns and responses to change. We will add the detailed information of the data source (spatial information of global SOC and WorldClim) in the revision and discuss the inherent uncertainties associated with those estimated k values.

---

## Author Response (AR1)

**Cover letter**

Dear Dr. Subke,

Included in this submission is our revised manuscript (bg-2017-338) entitled, "*Stable isotopic constraints on global soil organic carbon turnover*". We would like to thank both the editorial team and the reviewers for the time and effort they have put into assessing the previous version of this manuscript. Based the highly constructive critiques we received from the referees, we have made a number of changes to the manuscript that we feel have improved its quality immensely.

In the revised manuscript, we revised our dataset and corrected the errors according to soil profiles from savanna ecosystems and tropical forests. We added 20 new soil profiles into the revised dataset from 8 peer-reviewed articles, which resulted in 177 separate soil profiles in the revised dataset. We also added the detailed information of the data source (spatial information of global SOC and WorldClim) in the revision and discussed more on the inherent uncertainties associated with those estimated k values. The detailed reply to each comments from the reviewers was attached below.

We look forward to hearing from you in due time regarding our submission and to respond to any further questions and comment you or the reviewers may have.

Sincerely,

Edith Bai

Institute of Applied Ecology, Chinese Academy of Sciences,
No. 72 Wenhua Road,
Shenyang, Liaoning,
110016, China
Telephone: +86-24-83970570
Fax: +86-24-83970300
*E-mail: baie@iae.ac.cn

**Point by point response to reviewer**

**Reviewer 1**

A. SUMMARY OF THE REVIEW The topic is fully relevent for publication in Biogeoscience. The study confirms the correlations between $^{13}C$ enrichment down the depth in soil profiles and environmental variables, that bave been already reported by several authors on smaller datasets. The title makes sense. But the manuscript requires

significant revisions for two major reasons. - The first concern the consistency of the dataset itself, i.e., some erroneous attribution of data to biomes, omission of true tropical savannas profiles and profiles that reveal paleo C4-vegetations. The contribution of past C4 vegetation (delta $^{13}C$ -10 to 13‰ in such profiles is misunderstood. This leads to (i) non-representative values of beta and (ii) overestimation of temperature dependence of $^{13}C$ enrichments.

We agree that there are some erroneous attributions of data to biomes, omission of true tropical savannas profiles and profiles that reveal paleo C4-vegetations. We revised the dataset and corrected the erroneous attributions of data to biomes. We deleted the tropical savannas profiles and those profiles that reveal paleo C4-vegetations.

The second point concerns interpretation beyond this bias. The discussion tends to maximize the importance of kinetic fractionation by microbes as an explanation of beta's variance, but without providing results supporting it. I suggest reducing the section of discussion on this point. These comments are detailed in point "C. Extended review".

Thank you for your suggestion. We have reduced the discussion part on effect of kinetic fractionation on beta's variation and discussed the uncertainties of this method.

Finally, the "Stable isotopic constraints on global soil organic carbon turnover " is there, but has to be recalculated after database correction. Interpretation of the correlations in terms of processes should be minored. I suggest to correct the database by restrict it to pure C3 ecosystems, and furthermore make the proposition to merge this database with another one, therefore doubling the number of observations (see B. below).

B. ONE UNUSUAL PROPOSITION TO IMPROVE THIS STUDY. In line with the philosophy of Biogeoscience Discussion, which stimulates interactive and cooperative research more than competitive research, I make the proposition to provide 155 additional profiles worldwide. In the frame of the COST (European COoperation in Science and Technology) action SIBAE (Stable Isotopes in Biosphere-Atmosphere-Earth System Research; 2009-2013), a group of 10 scientists (10 institutions) has built and analyzed an exactly similar database of 196 World 13C/C profiles and beta values under pure C3 ecosystems. 43 are common to yours, 99 in other peer-reviewed articles, and 56 in non peer reviewed literature (Figure 1). This dataset shows significant multiple regressions with climate and clay, which are similar to those presented in the present manuscript BGD 2017-338, but with less predictive value. This less predictive value is in accordance with the above-mentioned biases, with more profiles exhibiting less negative beta values (including some positive) and more varied environments. If the authors of MS BG2017-338 accept this proposition, a final dataset merging the present dataset (after correction) with SIBAE's one, would provide a stronger view of the stable isotopic constraints on global

soil organic carbon turnover.

We revised the dataset and restricted it to pure C3 ecosystems. We deleted the soil profiles in tropical savanna and tropical forests which are marked by ancient C4 vegetations (see below). To compensate, we added 20 new soil profiles into the revised dataset from 8 peer-reviewed articles, which resulted in 177 separate soil profiles in the revised dataset-these studies were not available or missed when we first started gathering our data set. The addition of these additional data did not alter our summary conclusions, yet added robustness to our overall global findings. We thank the reviewer's suggestion to add 20 more profiles from an unpublished source; however, because our analysis is focused on synthesis of the published literature, we would not add this dataset, but would like to share our full dataset once it is published.

1. Consistency of the database.

Almost all so-called "tropical savannas" profiles in the database refer to afforestation or tree encroachments in C4 savannas. In published "real" tropical savannas profiles, delta13C decreases with depth, with values close to -12 -15 ‰ in surface and lower than - 20‰ in deep layers because of the presence of millenary-old forest-derived carbon. These have therefore POSITIVE beta values (of course non log transformable).

If not log transformed, they would draw the "tropical beta" toward the opposite direction. In the so-called "tropical savannas" (profiles # 85, 86, 87, 106-112), reforestation leads to a strong gradient from C3 signature in the top, and predominant C4 signature below. These systems were precisely chosen by the authors to analyse C dynamics through 13C signature change, and are not representative of world savannas. In some of the cited papers, profiles with positive beta were omitted. Interpretations of highest ln(-beta) in this "expansive and dynamic biome" (line 149) are therefore based on forest expansion data! As a result Figure 3 is wrong: either real savannas should be included (positive beta) or tropical savannas and these C4 to C3 conversions should be removed from the database. The latter is my suggestion.

We agree with reviewer comments and the "tropical savannas" profiles with number 9, 85, 86, 87, 93, 94, 106-112, 140 and 141 were removed from the revised dataset.

Beyond the case of "savannas", several profiles under tropical forests are marked by ancient C4 vegetations (profile numbers 78, 82-84, 115-121). They have been studied for this reason and are therefore not representative of world tropical forests. In both cases, beta is not linked to C turnover, but paleoclimate, as in many regions of the world. Almost all profiles with beta < -5 in the database are concerned. This overestimation of tropical ln(-beta) strongly affects Figure 4, the correlation with MAT or MAP (Figure 5), and Table 1, i.e., the main results.

We agree. The profiles of 78, 82-84, 115-121 were deleted from the revised dataset.

"C. Extended review"

2. Discussion of the relationships between beta and other variables (beyond paleo C4 vegetation)

The discussion might sometimes be confusing. A "kinetic isotope fractionation" associated to biodegradation process (decay) would not directly imply a dependence on the rate (speed) of decay, i.e., the turnover rate. In the Rayleigh distillation equation, beta is typically independent on the rate. A partial explanation of the variance in beta by the turnover rate rely on complex

processes (e.g., Acton, Garten, Schlesinger), and should not neglect other sources of 13C variations, such as the change in plant isotopic composition with time, post-photosynthetic fractionation in plants, bioturbation, isotope composition of nitrogenous compounds, etc., which can also be involved in the correlation of beta with carbon turnover rate. On the contrary, the discussion tends to minimize these processes (lines 215 to 234). Furthermore, the magnitude of a kinetic fractionation by heterotrophic respiration in soils is still debated (e.g., Breecker et al., 2015). Since the results provide no new demonstration, I suggest minoring this part of the discussion. Since the dataset includes turnover rate (k), some hypotheses of factors affecting the "turnover rate" (e.g., MAP > 3000 mm) might be discussed also on the basis of k, and not only ln(-beta). The apparent decrease of ln(-beta) under climates with MAP > 3000 mm is probably linked to overestimated -beta in moderately moist tropical areas (C4 zone).

Thank you so much for your suggestion. We added more discussion on the factors that affected soil 13C variation with depth beyond microbial fractionation. We also discussed the uncertainty when using beta value to study soil carbon cycling at the global scale.

3. Details. Precise when defining beta that you used the decimal logarithm ("log" is ambiguous). Table 1 and Figure 5 legend: indicate that N (%) and Clay (%) refer to topsoil. Table 1 Add one digit to the regressor of MAP. Figure S1 is after Acton et al. 2013

Thank you for catching those and the errors mentioned above were corrected in the revised manuscript.

**Reviewer 2**

The manuscript "Stable isotopic constraints on global soil organic carbon turnover" by Wang et al. presents an interesting approach of deriving information about SOC decomposition kinetics from stable carbon isotope information along the soil profile. For this, they derive a slope "beta" from the relationship of $\delta^{13}C$ values and SOC content of soil profiles across the globe, and then relate these "beta" values to calculated decomposition kinetic constants "k" (or more precisely their log-transformed negative values). They state that the highly significant linear relationship of the log-transformed variables can be used to derive SOC decomposition kinetics from $\delta^{13}C$ profiles of SOC. Furthermore, they relate these ln(-beta) values to four different parameters, i.e. MAT, MAP, soil clay and nitrogen content. For all four parameters they find significant relationships with ln(-beta). This approach is interesting and might be promising if proven to be reliable.

The weak part is the calculation of the kinetic decomposition constants with several secondary data sources and a fixed relationship between heterotrophic and total soil respiration, which might be too much of a simplification for this global approach, given the large range of ratios between heterotrophic and autotrophic respiration found for different ecosystems and conditions.

1) The kinetic decomposition constants k for the different soil profiles have been calculated by assuming steady-state conditions, i.e. SOC input and output are in equilibrium. While this assumption might hold true for many of the sites, there is no evidence provided that this really is the case.

2) The SOC stocks, which represent the denominator in equation 1, were extracted from the Global Organic Soil Carbon and Nitrogen (Zinke et al. 1998). There is no mention whether there was an exact match between the soil profiles used in the present study, or whether spatial approximations were made, and if yes, which criteria were used for these spatial approximations.

3) Heterotrophic soil respiration was calculated from total soil respiration by a fixed linear relationship adopted from Bond-Lamberty et al. (2004). Given the large variability of the fraction of Rh to total soil respiration (varying between 10% and 90% in vegetated ecosystems), this approach is highly questionable.

4) Also total soil respiration was not measured, but derived from a climate-driven regression model (Raich et al. 2002).

5) And finally, climate data were derived from WorldClim as a function of latitude and longitude (what about altitude?), whenever climate data were not available in the literature tapped in this study. Again, there is no mention whether there was an exact match between the locations of the present study, or whether spatial approximations were made, and if yes, which criteria were used for these spatial approximations.

Given all above-mentioned uncertainties concerning the calculation of the key variable of the study, i.e., the SOC decomposition rate constant k – which by the way is an apparent constant, as it is a composite of the decomposition of several SOC pools with different decomposability/recalcitrance – the reader would expect an extensive uncertainty analysis. However, not a single attempt was made to quantify those uncertainties, which certainly will amount to a large relative error due to multiple convolutions of single functions and error propagation. Also no mention is made of this crucial point in the discussion, and how this might affect the far-reaching conclusions drawn.

**Authors' response:** We would like to thank the reviewer for the time and effort of into reviewing our manuscript. We agree with the reviewer's comments that there are uncertainties associated with kinetic decomposition constant k – which have been discussed previously. However, to our knowledge, the approach we took is the best available for providing integrative soil carbon decomposition rates estimates along profiles. Numerous published modeling studies have used approaches involving multiple data sources and assumptions – similar to our approach. While we agree that the coupling of different data sources inherently injects uncertainties – and we have discussed the caveats in revision. Getting quantitative with the uncertainty is unfortunately not possible (as other studies have noted), for the following reasons. First, we mainly focus on soil 13C-based proxy and its variations with MAT and MAP in this manuscript. Hence, we compared our beta value with the kinetic decomposition constant k to explore correlation between these two factors, but we can't quantitatively assess the relationships at the global scale. This does not devalue the correlations we find across sites, though it does limit our quantitative assessment, pointing to an area for future research for the community. Thus, if useful, we could move the comparison between beta and k to the supporting information if this seems like the best approach.

Second, just because there are large uncertainties with the kinetic decomposition constant k, we believe the independent approach provided in this manuscript using carbon isotope variations along soil profiles is a promising approach. Indeed, it allows for larger-scale geographic exploration of soil carbon decomposition at the global scale in a way that differs fundamentally from current approach. We believe that this constraint can be used to help benchmark global models, which are lacking in their ability to generate global soil C patterns and responses to change.

We added the detailed information of the data source (spatial information of global SOC and WorldClim) in the revision and discussed the inherent uncertainties associated with those estimated k values. We added the following discussion in the revised manuscript.

*"The SOC stock for each soil profile was extracted from a global soil organic carbon map (within 1 m depth), which was created by IGBP-DIS (1998) with a resolution of 0.5 by 0.5 degree; and the mean annual soil total respiration (Rs) was extracted from a long-term dataset with a resolution of 0.5 by 0.5 degree (Raich et al., 2002). We added the two dataset and the coordinate of soil profiles into ArcGIS (version 10.0, ESRI, Redlands, CA) to extract SOC stock and respiration of each profile using spatial Analysis tool."*

[revised manuscript text omitted]

---

## Author Response (AR2)

**Response to Editor Comments:**

76: Supplemental table S1 has 177 entries, which matches your statement in the abstract, but not what is being said here. Further, entry 196 in the supplemental table lacks results, so is seemingly not included? (I.e. should total number of profiles be 176?)

We thank you for catching this error. The number '196' soil profile has been deleted in the supplemental table. So, the final dataset included 176 soil profiles, and we have corrected this in the abstract (Line 12) and the text (Line 76, 82, 89, 127 and 391).

103-105 Replace "We applied […] millennial time-scales" by: "In order to calculate the SOM decomposition rate constant, we assumed that input of organic matter to soil and decomposition are in equilibrium with no change in SOM stocks over time (steady state assumption). We acknowledge that this assumption is not valid in most disturbed environments, including agricultural systems, but it provides a reasonable approximation in natural ecosystems, where SOC turnover has equilibrated on century to millennial time-scales." (This is a suggestion from me, please feel free to rephrase. I think that simply stating "steady state assumption" was not clear enough and that a slightly more elaborate justification was needed.)

Thank you very much. We agree with your suggestion and have replaced the sentence in the revision (Line 106-109).

228-234: This section is important, but it is poorly integrated into the overall discussion. These points reflect the concerns of referee 1 well, but the reader is left not knowing how significant these uncertainties are for your findings. I suggest moving this section up into the main part of the discussion and qualifying whether you think that the patterns you observe in your study are robust despite these limitations. I'm not asking for a lengthy elaboration of implications, but a clearer indication of whether, in your judgement, these limitations are a major limitation to your conclusions.

Thank you for your suggestion. We moved this part up into the method (Line 85-87) and discussion (Line 208-209 and 213-214).

Figure 4: In panel B, the axes are seemingly labelled incorrectly. To match data in panel A, k seems to be scaled on the y-axis, and ln(-beta) on the x-axis, but your axis labelling suggests the opposite.

Thank you. We have changed the axes.

**Marked-up manuscript version**

[revised manuscript text omitted]

---

## Author Response (AR3)

**Response to Editor Comments:**

Number of soil profiles included is stated as "149" in line 76, contradicting other occasions in the manuscript and the author's response. Please amend to read "176".

**Author's response:** The total number of soil profiles in our dataset is 176, which includes 149 profiles that were compiled from previous studies (Line 76) and 27 profiles that were from a grassland transect study conducted by our group (Line 88) (Wang et al. 2017). We describe the number of soil profiles in line 89.

[revised manuscript text omitted]